# Joining the Dots—Understanding the Value Generation of Creative Networks for Sustainability in Local Creative Ecosystems

Marlen Komorowski, Ruxandra Lupu *, Sara Pepper and Justin Lewis

College of Arts, Humanities and Social Sciences, School of Journalism, Media and Culture, Cardiff University, Cardiff CF10 1FS, UK; KomorowskiM@cardiff.ac.uk (M.K.); PepperS1@cardiff.ac.uk (S.P.); LewisJ2@cardiff.ac.uk (J.L.)
* Correspondence: LupuR@cardiff.ac.uk

**Abstract:** In recent years, the ecological shift from an economically driven model of arts and culture to that of an ecosystem in the creative industries determined the emergence of a range of new bottom-up, place-based networks herewith referred to as "creative networks". This article explores how these networks can generate sustainability for local creative ecosystems through a value network approach. Building on the quadruple helix model to identify the actors in these networks, this study explores the relationships and value flows between the actors of 22 identified creative networks across the UK. It then maps these relationships using data gathered through a mixed methodology that includes survey data and focus group research. Our findings show that creative networks operate as central nodes of the local creative ecosystem, functioning as a 'glue' inside the otherwise very heterogenous creative industries. From this position, creative networks can act as catalysts for sustainability. However, the economic, cultural, and social value created by creative networks is often overshadowed by other challenges including a lack of funding and a lack of understanding from policy makers or the public.

**Keywords:** creative networks; value networks; creative industries; quadruple helix; creative ecosystems; sustainability

## 1. Introduction

Research into creativity has progressed significantly in the past few decades from individual to collective, and more recently to a systems-based account of creativity [1–3], which looks at the whole creative ecosystem. We define creative ecosystems as spatial agglomerations of creative activities in which links/collaborations between creative actors take place [4]. Such a system perspective [5] addresses the different contexts and levels at which creativity occurs—individual, group, institutional and sociocultural [6,7]. Research on creative ecosystems [8–13] builds on the understanding that creativity is a phenomenon that arises in dynamic transactions between individuals. The proposed idea that creativity is distributed [14–16], scalable [17,18] and networked [19,20] emerged from an understanding of the open, fluid and dynamic context within which creativity occurs. We argue, that by advancing research into networks in which creativity can flourish [21], this approach to creativity gives room to look at the whole creative ecosystem [22] and how this can create sustainability.

Hawkins describes in this context networks of habitats both intangible and physical spaces whose potential can be transformative if they prioritize resilience and sustainability [23]. Spaces, such as cities and regions, have played an important role in the process of rethinking and understanding creative ecosystems and their networks. Most prominently, the concept of the "creative city" has created a new way of thinking about how important the geography of the creative industries is [24–26]. Furthermore, in recent years,

research has made further advancements by applying a more systems-based approach to the study of creative cities [27,28], which determined a shift in attention from the cultural infrastructure and aesthetic dimension of cities towards a focus on city networks and interactions [29].

Because of the dynamic and fluid nature of the creative industries [30] and its specific structure that distinguishes creative sectors from other industries, we can find that networked forms of creative organisation emerged in the last decades [31–33]. Whilst some versions of these organisations can be found in most sectors, the creative industries, which is based on clusters of small enterprises and a large freelance workforce, has seen the emergence of a more specific type of organisation with specific characteristics, here-after referred to as "creative networks".

Creative networks are not new to cities and regions in the UK. Nevertheless, a range of new initiatives established in recent years point to an increasing interest in and need for a place-based type of organisation that creates sustainability by offering a range of support and enabling services to the local creative industries. This article explores creative networks as an increasingly important phenomenon for sustainability in creative ecosystems, with the creative industries being one of the largest growth contributors to the local economy [34,35], and whose recognition has led to an increase in support and policy commitments.

Despite the growth in the number and scale of creative network organisations over the past 10 to 20 years, there is a significant gap in both academic and policy analysis and understanding. Research into the nature of creative networks [36–39], frameworks for analyzing these [40,41] as well as spill-over effects [42,43] point to the need to advance our understanding of how creative networks function and create value for the creative industries. This research maps the similarities and differences among UK creative networks and uses value network analysis based on the quadruple helix model to explore how creative networks generate added value. We concentrated an increased focus on the intangible 'soft' values that creative networks create associated with economic development. Understanding the functioning of creative networks can help us better grasp the unique character traits of the needs of the creative industries and the way they derive competitive advantage [44] in places. Exploring the added value of creative networks allows us to grasp the link between creativity and urban sustainability and develop more sophisticated forms of measurement of the performance of creative actors in cities and regions in the future. In summary, our study aims to bridge the current gap in research concerning methods for capturing the benefits of creative networks for territorial sustainability [45–47].

The article targets practitioners, researchers and policymakers and is structured in three parts. In the first part, we introduce the research background, contextualize creative networks as an object of research and discuss the methodological framework for analysis. In the second part, we discuss research findings, exploring the similarities and differences between creative networks and detail their roles in value generation. In the last section, we discuss the findings and develop recommendations for the future.

## 2. Materials and Methods

### 2.1. Research Background

The creative industries are a key growth driver for the UK economy. Recent figures published by the Department for Culture, Media and Sport show that the UK's creative industries contributed more than £111 billion to the UK economy in 2018 [48], growing more than five times faster than the economy as a whole [49]. This equates to a contribution of almost £13 million to the UK economy every hour and the creation of nearly 2.6 million jobs [50].

In 2018, the UK Government's Industrial Strategy recognized that the creative industries are an "undoubted" national economic strength and committed to a sector deal, aimed at putting the UK at the forefront of emerging technologies through support to creative industries. Additionally, to address the gaps in prosperity between London and the Southeast and the rest of the UK, new local initiatives emerged. These place-based policy

approaches led to increasing awareness of the importance of existing creative networks for local economic sustainability, and in some cases to policy efforts towards the creation of new ones.

Academic and policy research acknowledges the importance of creative networks and their contribution to sustaining and developing the creative industries [51]. The NESTA Manifesto for the Creative Economy (2013), for example, highlights the importance and value of "building new peer networks where none exist and supporting those that do exist" as one way of supporting the creative industries. Their rising importance is a response to the highly dispersed make-up of the creative industries, characterized by a high number of small companies, freelancers and self-employed workers, few large corporate structures, a reliance on project-based/contractual work and the important role of social dynamics. This makes the creative industries less stable and sustainable, prone to being highly impacted by changing economic circumstances and crises. The creative economy, due to the lack of large organisations working in these sectors, does not have the capacity itself to create the necessary networks to generate more sustainability. By forging relationships of trust [52], networks stimulate knowledge creation and industry development, giving rise to a culture of collaboration in which risks associated with economic precarity and the dispersed nature of the creative workforce are minimized [53]. In summary, we argue that creative networks are creating more sustainability for local creative industries, which in turn makes local economies more sustainable in the long run.

### 2.2. Creative Networks as an Object of Research

Academic and policy work considers the make-up of "creative networks" in a variety of different ways. Concepts related to, but by no means interchangeable [54] with, creative networks include creative cities [55], creative ecosystems [56], creative knowledge exchange hubs [57], creative clusters or agglomerations [58–60], cultural hubs [61], communities of practice [62,63], social networks [64], creative places [65] and (co)-creative living labs [66]. This variety of circulated concepts underpins the broad literature that exists where two strands of thought prevail [67]: A dominant strand that explores the agglomeration and clustering of creative industries (co-location understanding [68]) and a second and more recent one that analyzes creative industries from a network and connectivity perspective (network understanding). So far, there is little communication between these two strands of research, and capturing the interconnections between the economic and cultural/relational dimensions of networks remains particularly challenging inside the creative industries [69].

We do, however, begin to see some overlap in the research on creative collaboration. The " networked flow" model [70] is a good example of new methodological approaches embracing the complex nature of creative networks. Another example is the growing research in innovation management around living labs, a model of innovation ecosystem characterized by an open, inclusive and co-creative approach. Whether methodological or ontological, this research is underpinned by the idea that creative networks create benefits for creative workers and organisations in the form of capacity-building and/or new partnerships/work relations, which creates more sustainability in the industry. This acknowledges networks as a collection of "actors" and their strategic links with each other [71].

In this article, we focus on creative networks as network organisations (sometimes as their own legal entity, sometimes as part of another organisation), that have dedicated people working to create collaboration and/or growth/sustainability in the local/regional creative industries. We have found a recent trend of cities and regions establishing such organisations, which have similar characteristics. Consequently, creative networks in our study are all:

- City/regional network organisations;
- Working with (multiple) creative industries sectors;
- Rooted in and working for a city or region;
- In operation for a minimum of 1 year (for analysis purposes);
- Working for both creative individuals and organisations.

These criteria incorporate a variety of network organisations we identified that operate in the UK and that represent a recognizable cohort of like-minded organisations, which have received, in recent years, increasing attention through policy makers. Our focus is on place-based initiatives with the aim to bring people together in real time in real places to make their local creative industries more sustainable. Below, we give more details about their specific characteristics and commonalities. We do exclude broader sectoral networks, associations (e.g., the Royal Institute of British Architects) or agencies with national remits (e.g., Creative England, Creative Scotland, Creative Wales or UK Arts Councils) to be able to focus on this new phenomenon of creative networks that have been established in recent years.

*2.3. Methodology*

In our study, we adopt value network analysis as a core methodological framework. Value network analysis draws from a theory based on living systems, knowledge management, complexity theory, system dynamics and intangible asset management [72]. It relies on the visual representation of flows and actors in networks, thereby facilitating the identification of types of created links and actors and establishing a typology of value creation. Compared to other value-mapping models such as process views, social network analysis or clustering techniques, value network analysis enables us to map the distinctive features of creative networks in the creative industries.

These features come from the mode of organisation of the production of creative products and services that breaks down the traditional analytical divisions between public (traditionally cultural organisations receiving public subsidy) and private (commercial creative companies), formal and informal and for- and not-for-profit activities [73–76]. In doing so, we follow the quadruple-helix model, which recognizes four major actors in the system: Science, policy, industry and society [77]. As an analytical tool that shares characteristics of complex systems [78], it helps to identify the distinct actors of the network that we aim to analyze. This is important for a network visualization process that pays close attention to the distinctive value generation dynamics in creative industries, creating a value network with the creative networks as a connector that builds on a specific value flow. The quadruple-helix model helps to organize the possible actors and flows between them in the value network to identify their connectedness. This can reveal both strengths and gaps that can guide creative network growth and sustainability and best practice for the creative networks themselves, as well as for policy makers and practitioners. The goal is to "join the dots" of the creative networks' value creation.

Building on this framework, we applied a mixed-method approach including desk research, surveys and the organisation of a focus group. In the first instance, our definition of creative networks informed desk research to identify creative networks in the UK. We then invited these networks to participate in a survey in order to more closely map how, why and where these networks existed and to assess their differences and similarities based on 6 criteria (inspired by an innovation ecosystem framework [79,80]): (1) Location and scale where the creative network operates; (2) scope and scale of the creative network; (3) development (path-dependency) of the creative network; (4) organisational structure of the creative network; (5) services and ambitions of the creative network; (6) challenges of the creative network.

Data were collected from the 22 UK creative networks identified through desk research; 15 of these then took part in the survey (from July–September 2020 via Qualtrics), and then participated in a subsequent focus group aimed at gathering in-depth data (via the Miro tool) about the functioning of the networks and co-creating a new framework for understanding the process of value generation of creative networks for sustainability.

## 3. Results

### 3.1. Commonalities and Differences between Creative Networks

During the mapping phase, we identified 22 creative networks in the UK, covering 21 cities/regions (Figure 1) (Bristol has two networks that fit our definition), constituting a representative but not exhaustive list (Table 1). We then mapped the selected creative networks against the six established criteria.

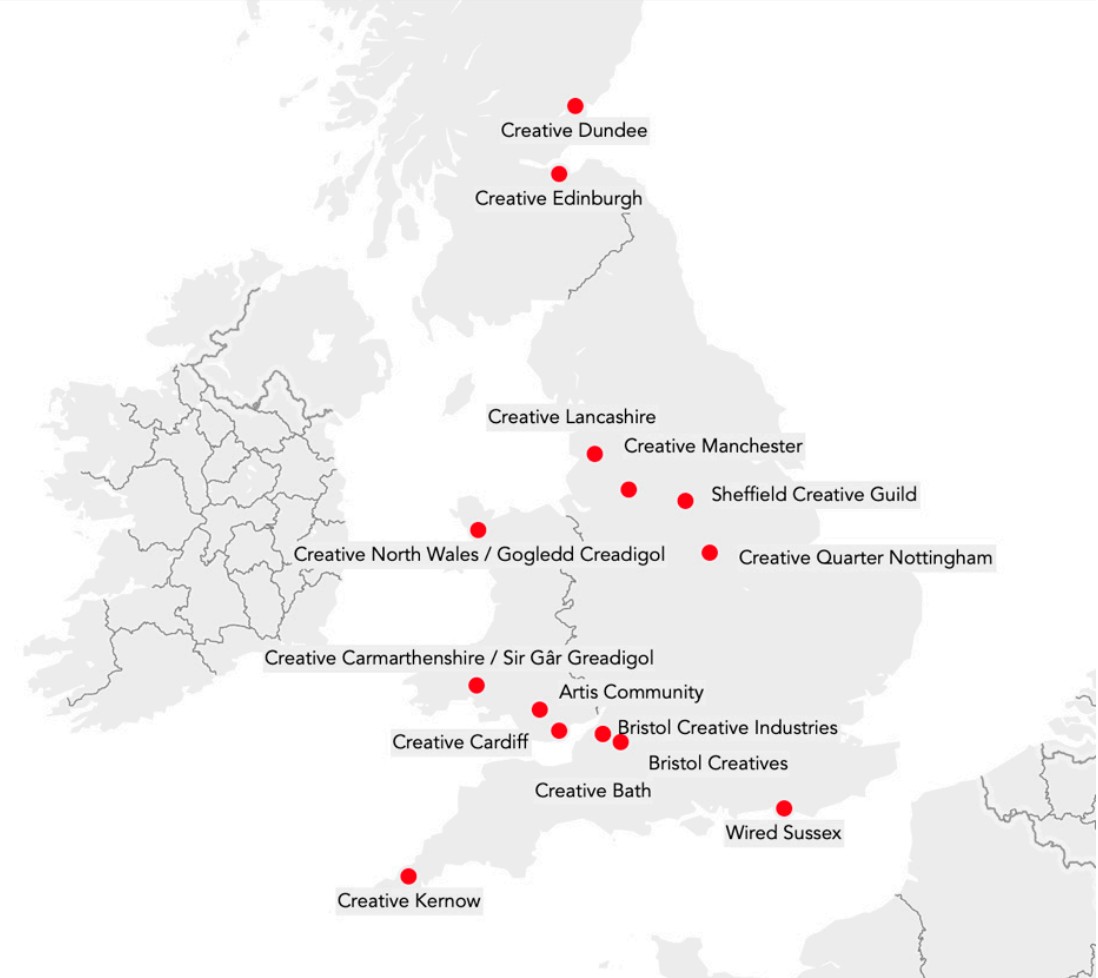

**Figure 1.** Geographical distribution of surveyed creative networks.

Our findings show that most creative networks (71%) serve the broad creative industries and most of them (65%) are present in large urban areas. These networks mostly operate at the city (47%) or regional level (40%) and most use a common nomenclature indicating a commonality of purpose (Creative Bath, Creative Cardiff, Creative Clyde, etc.). In terms of size, networks vary significantly: Around half (47%) are small with less than 500 members and a third (33%) are large with more than 1000 members. Collectively, we estimate that these networks provide access to about 6% of all creative businesses in the UK, turning them into important local and regional outreach organisations (25% of these networks claimed to reach up to 90% of creative companies/organisations in their city/region).

**Table 1.** Overview of identified creative networks.

| Name of Creative Network | Location | Network Scope | Network Size (Number of Organisations/Freelancers Involved) | Year Founded | Interview Attendance | Survey Respondent |
|---|---|---|---|---|---|---|
| Artis Community | Rhondda Cynnon Taff | Creative industries | 100–200 | 2018 | ✓ | ✓ |
| Bristol Creatives | Bristol | Visual & applied artists | 900–1000 | 2006 | ✓ | ✓ |
| Bristol Creative Industries | Bristol | Creative industries | 500–600 | 2005 | ✓ | ✓ |
| Creative Bath | Bath | Creative industries | 600–700 | 2008 | - | ✓ |
| Creative Cardiff | Cardiff | Creative industries | 3000–4000 | 2015 | ✓ | ✓ |
| Creative Carmarthenshire | Carmarthenshire | Film/video, radio/television & music | 50–100 | 2018 | ✓ | ✓ |
| Creative Clyde | Glasgow, Scotland | n.a. | n.a. | n.a. | - | - |
| Creative Dundee | Dundee | Creative industries | 200–300 | 2013 | ✓ | ✓ |
| Creative Edinburgh | Edinburgh, Scotland | Creative industries | 4000–5000 | 2001 | ✓ | ✓ |
| Creative Gloucestershire | Gloucestershire | n.a. | n.a. | n.a. | - | - |
| Creative Kernow | Redruth | Creative industries | 2000–3000 | 1995 | ✓ | ✓ |
| Creative Lancashire | Lancashire | Creative industries | 2000–3000 | 2004 | - | ✓ |
| Creative Leicestershire | Leicestershire | Creative industries | n.a. | n.a. | ✓ | - |
| Creative Manchester | Manchester | Creative industries | 100–200 | 2018 | ✓ | ✓ |
| Creative North Wales | Caernarfon (West Wales) | Digital creative | 100–200 | 2012 | ✓ | ✓ |
| Creative Quarter Nottingham | Nottingham | Creative industries | 200–300 | 2012 | ✓ | ✓ |
| Creative Stirling | Stirling | Creative industries | n.a. | n.a. | - | - |
| Creative Swindon | Swindon | n.a. | n.a. | n.a. | - | - |
| Culture Central | West Midlands/Coventry | Arts & culture | n.a. | n.a. | ✓ | - |
| Culture Northern Ireland | Derry/Londonderry | n.a. | n.a. | n.a. | - | - |
| Sheffield Creative Guild | Sheffield | Creative industries | 700–800 | 2016 | ✓ | ✓ |
| Wired Sussex | Brighton | Media & createch | 1500–2000 | 2007 | ✓ | ✓ |
| Total identified: 22 creative networks | Number of cities: 21 | | Total members: >18,000 | | 15 | 15 |

In terms of development, we identified two growth periods for these networks (2004–2008 and 2015–2018), the latter being the most prolific. While most networks (40%) were founded with government/local authority involvement, other stakeholders such as companies, universities and individuals are also involved in establishing these networks. In terms of structure, networks are like start-ups or micro-businesses, both in terms of organisational structure (80% of networks are non-profit) and employment capacity (50%

of networks employ 2–5 full-time staff members). The size and structure of these networks are related to their funding mechanisms: smaller networks tend to rely on public funding, while larger ones have greater diversity in their incoming streams to include other sources such as membership fees. The fact that most of the networks face challenges such as a lack of resources (time/money) and external understanding renders them specifically vulnerable to risks such as those posed by COVID-19 or economic crisis. Despite their structural differences, networks share two main goals: To strengthen collaboration and create awareness of and promote their local creative industries, thereby making the local creative industries more sustainable.

### 3.2. Value Network Mapping

After mapping the 22 creative networks against the six criteria, we circulated surveys and organized a focus group with representatives of the creative networks to better explore how these networks generate value, the results of which we summed up visually in Figure 2. Figure 2 follows the quadruple-helix model using a value network analysis approach to organize and display the four types of actors that are linked by creative networks, which are highlighted in the different boxes:

A. Government, which includes both UK government agencies and bodies as well as regional and local government agencies and bodies;
B. Academia, which includes education institutions and research centers;
C. Civil society, which includes charities, NGOs, associations and other groups and organisations relevant for the creative industries;
D. Industry, which includes both organisations in different sizes and freelancers from the creative industries and the wider economy.

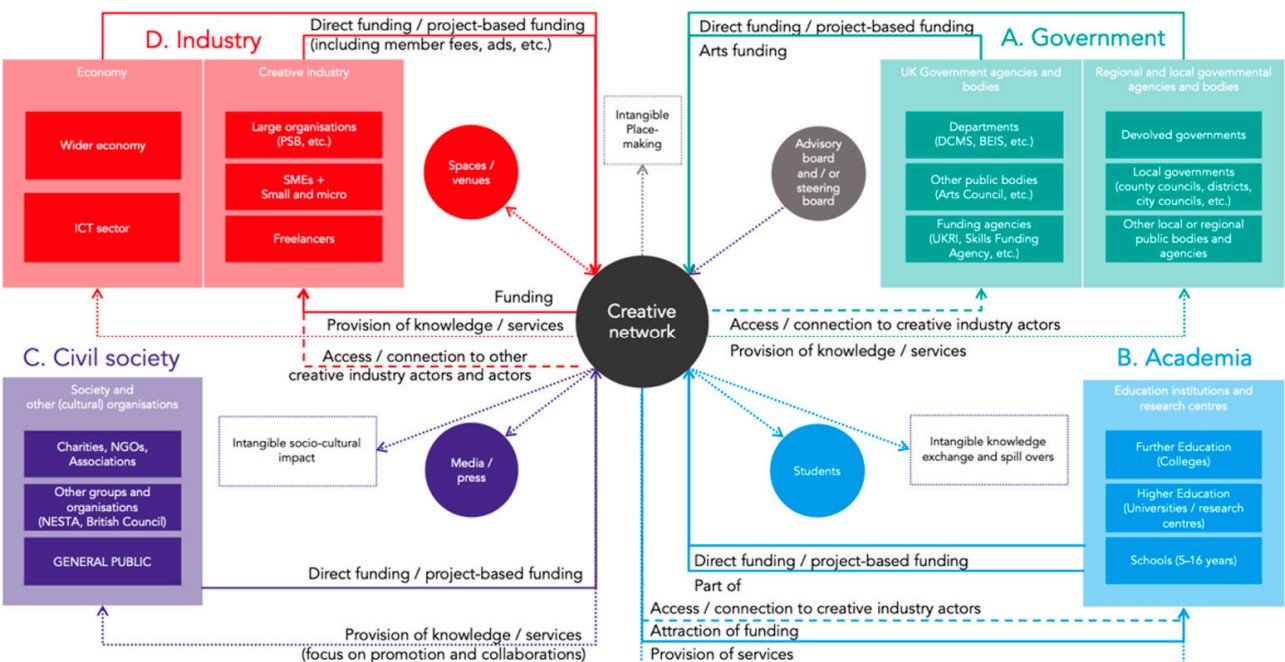

**Figure 2.** Value network of creative networks.

We also highlight specific sub-groups of these actors that we have found to have a specific impact on the creative network's value network (as indicated in the circles), namely the advisory or steering board of the value network, which is often made up of representatives of all four actor groups, students of education institutions, media and press, which are often involved in creative networks' activities and spaces and venues, which highlights the importance of localities of creative industries' activities in cities and regions.

The creative network links these actors together through specific value flows both directed from and to the creative network as indicated by the arrows. We found that creative networks operate on a continuum between economic and (often less tangible) cultural value. We have categorized these value flows into four categories, which are highlighted in Figure 2 through the different types of arrows used:

1. Monetary flows, which integrate a different kind of (or attraction of) direct and project-based funding;
2. Collaboration/cooperation flows, which includes how creative networks create access to and connections between different actors in the value network;
3. Service/knowledge flows, which includes the provision of services and other knowledge to different actors;
4. Less tangible cultural value flows, including the creation of values through place-making, socio-cultural impacts and spill-over effects.

We argue that these value flows that are generated through creative networks in local creative industries create sustainability within the local creative ecosystem. We outline how the actors are involved in the value network and how value flows are generated in more detail below, using case studies for illustration.

### 3.3. Value Network Actors

Looking at the value network actors in creative networks enables us to delve deeper into micro aspects of the relationships and thus enhance our understanding of dynamic relations between these groups and how this can create sustainability [81].

### 3.3.1. The Government and Creative Networks

Devolved and local governments in the UK are essential actors in the value network of creative networks. These bodies fund activities of creative networks (direct/project-based funding). In exchange, these networks provide governments with knowledge and services, as well as access to other actors in the network. A good example of this is the agreement between Creative Manchester and the Manchester City Council, through which students at Manchester University can work every year on placement at the Manchester International Festival (https://www.youtube.com/watch?v=2-5bgKcCKo0, accessed on 20 September 2021). Such initiatives show how creative networks create both economic and cultural value while enforcing sustainability in the local economy. Policy makers often use creative networks as representatives for the local creative industries, insofar as they generate knowledge exchange and feedback loops between the government and creative industry players.

> Case study 1—Creative Edinburgh partners with Creative Informatics: Alongside the University of Edinburgh, Edinburgh Napier University and CodeBase, Creative Edinburgh are partners in Creative Informatics (https://creativeinformatics.org, accessed on 20 September 2021). Through five key funding programs and regular events, Creative Informatics enables creative individuals and organisations to explore how data-driven technologies can enhance their work. It is funded by the Creative Industries Clusters Programme managed by the Arts and Humanities Research Council as part of the Industrial Strategy, with additional support from the Scottish Funding Council (https://www.creative-edinburgh.com, accessed on 20 September 2021).

### 3.3.2. Academia and Creative Networks

Academic institutions can also be a catalyst for the sustainability of creative networks through direct and project-based funding. In exchange, these networks can connect businesses with researchers with a specific shared research interest. This enhances collaboration around research, innovation, and skills, with creative networks offering connections to students and providing work experience for students and skills development for creative companies (e.g., in 2018, Wired Sussex partnered with the University

of Sussex on the 'Shaping sound like we do with light' project to sponsor a residency for a PhD student, https://www.sussex.ac.uk/study/fees-funding/phd-funding/view /962-PhD-Studentship:-Shaping-sound-like-we-do-with-light, accessed on 20 September 2021). Creative networks can also attract funding for the university (e.g., there are several research and engagement projects delivered by Creative Cardiff for the British Council, https://www.cardiff.ac.uk/__data/assets/pdf_file/0005/528872/Mapping-Ca rdiffs-Creative-Economy-English.pdf, accessed on 20 September 2021).

> Case study 2—Creative Cardiff (https://creativecardiff.org.uk, accessed on 20 September 2021) building the Creative Economy Unit at Cardiff University: Creative Cardiff as part of Cardiff University has contributed to the development of the Creative Economy Unit (CEU) at the university. The initial remit of the CEU was to develop research-led engagement. It began by creating networks of researchers within the University and researching existing city/regional creative networks (in places such as Bristol, Brighton, Edinburgh), building upon best practices in the UK and mapping the creative economy in Cardiff (https://www. cardiff.ac.uk/creative-economy, accessed on 20 September 2021).

### 3.3.3. Civil Society and Creative Networks

Charities, not-for-profit organisations, NGOs, associations and other organisations such as NESTA or the British Council benefit from partnerships and services through engagement with creative networks. For example, the British Council partnered with Creative Edinburgh and six other European hubs to deliver 'European Creative Hubs Network', a 2-year project that helps creative hubs connect and collaborate across Europe (https://www.creativeeuropeuk.eu/funded-projects/european-network-creative-hubs, accessed on 20 September 2021). Most creative networks are also highly engaged with the media and press to engage the wider public. Creative networks often promote creative projects or the local creative industries (e.g., through their web presence and PR and communications work); in turn, they can be funded by civil society organisations through programs such as those of the British Council or NESTA.

### 3.3.4. Industry and Creative Networks

Creative networks engage with a range of businesses and freelancers from the creative industries, with small- or micro-business relations dominating these connections. In some cases, organisations in other sectors engage in value generation in creative networks. As a result, the impact of value generated by creative networks refers not only to the creative industries, but also the wider economy. Creative Cardiff, for example, established the Coworking Collective, a partnership with several coworking spaces that shows the impact on spaces and places including activities outside of creative industries. Research also shows that the combination of creative workers and ICT leads to higher levels of GVA growth [82] (https://bighouse.org.uk/, https://creativecardiff.org.uk/research-and-projects/project s/immersive-south-wales, accessed on 20 September 2021) Finally, it is important to stress the role that creative networks play in engaging freelancers who often lack the capabilities to network in the same way that larger organisations do.

> Caste study 3—Creative Dundee's Amps programme (https://creativedundee.c om/amps-network/, accessed on 20 September 2021). The community members of Amps meet regularly, online and offline, to share news and ideas, discuss current issues and collectively build the future of the city. The Amps network offers an opportunity to join events designed to build connections, showcase local projects and develop collaborations throughout Dundee and beyond. Network members are eligible for the Community Ideas Fund, funded through membership fees (50% of subscriptions go towards the fund, the other 50% goes towards commissioning creative work).

### 3.4. Value Networks' Value Flows

In this section, we discuss the value flows generated by the types of relationships we have identified in more detail. While our study maps all types of identified value flows, it is worth highlighting the importance of intangible values on the sustainable development of territories. These values are connected to socio-cultural or behavioral models and play an important role in bridging the gap between corporate responsibility, stakeholder theory and sustainability [83].

#### 3.4.1. Monetary Value Flows

Securing funding is the most pressing challenge creative networks face. Our survey revealed that 4 in 10 creative networks rely on public funding as the main income source, supplemented by project-related funding, membership fees and other earned income (e.g., advertising). This kind of funding can come from all actors in the network, including government, academia, civil society organisations and industry. Income sources also correlate with the size and age of creative networks; the more established the creative network, the bigger the share of earned income, suggesting that, given time, networks can become more sustainable. At the same time, creative networks generate direct monetary value for other network actors through the attraction of project funding to universities as well as distributing direct network funding and the intermediation of funds. The membership scheme of Bristol Creative Industries presents a good example of this flow.

> Case study 4—Bristol Creative Industries membership: Bristol Creative Industries (https://bristolcreativeindustries.com, accessed on 20 September 2021) offers tiered membership based at different rates. At £45/year, they offer the Individual and Start Up rate for freelancers, small businesses and start-ups turning over less than £150k. For established and growing businesses and agencies they offer a business rate of £120/year. Student membership is free. Benefits include the ability to showcase businesses through the member directory, to participate in industry events, post jobs for free and to self-publish news and discounts at local retailers (https://bristolcreativeindustries.com/join/#Individual-startup-membership, accessed on 20 September 2021).

#### 3.4.2. Collaboration and Cooperation Value Flows

Research indicates that collaboration between creative businesses leads to more efficient use of knowledge [84]. Bringing people together in various ways is intrinsic to the nature and functionality of creative networks. Collaboration and cooperation thereby constitute a distinct value flow. This is also evidenced in our survey, with 9 out of 10 networks offering networking events, such as Creative Edinburgh's Mentoring Scheme (https://creative-edinburgh.com/what-we-do/mentoring, accessed on 20 September 2021). Currently, identifying and measuring the direct effect of the activities of creative networks on collaboration and cooperation is difficult, and addressing this methodological issue will be an important part of establishing their value.

> Case study 5—Creative Kernow's Cultivate scheme (https://www.creativekernow.org.uk/cultivator/, accessed on 20 September 2021): Creative Kernow's Cultivate scheme is a creative business support scheme. It includes a range of support measures for creative businesses with and through partners including one-on-one business advice, creative investment grants, internship incentives, specialist mentoring, skills development grants and a creative export program. They have a team of advisors who have an in-depth understanding of different segments of the creative industries as well as the knowledge and experience to support and cooperate with creative businesses (https://cultivatorcornwall.org.uk/, accessed on 20 September 2021).

### 3.4.3. Service and Knowledge Value Flows

Creative networks offer a range of different services (e.g., research and development, mentoring/incubation programs, access to equipment, studio space, hot-desking or co-working space, crowdsourcing [85]) and, as such, contribute to service and knowledge value flows. This knowledge can be particularly useful for new industry entrants.

Case study 6—Wired Sussex talent manifesto (https://www.wiredsussex.com/blogpost/1284027/skills-and-talent-manifesto-whats-happened-so-far, accessed on 20 September 2021): The Wired Sussex talent manifesto is a collective commitment that supports the network's goal of making the region the best place in the UK for anyone to build a digital career. This manifesto offers a collective commitment that businesses can sign up to, to support the aspiration to make Greater Brighton the best place in the UK to work in the digital sector resulting in a Pledge and a Diversity and Inclusion toolkit that members get access to (https://www.wiredsussex.com/initiative/1310656/skills-talent-and-diversity, accessed on 20 September 2021).

### 3.4.4. Other Intangible Value Flows

As already mentioned above, besides the more formal ways of value generation, creative networks foster other intangible values, which differentiate them from the classical value network of firms and other industries. These intangible (or less tangible) values cannot be captured as easily through the measurement of monetary, goods and service flows. They are deeply connected to the construction of place and socio-cultural identity. So, for example, we found that creative networks can build a sense of belonging and local identity through cultural value flows or the creation of social spaces. This can be articulated either as a way of generating spill-over effects through the overflow of concepts, ideas, skills, knowledge, and different types of capital [86] or as the contribution to the symbolic value of place identification [87,88], where tacit knowledge and relationship building are developed through personal encounters [89]. While these intangible values have been explored in research, our research suggests that creative networks play an important role in this form of value generation.

Case study 7—Portsmouth Creates' 'We Believe' arts trail (https://www.portsmouthcreates.co.uk/we-believe/, accessed on 20 September 2021): The 'We Believe' arts trail of Portsmouth Creatives was developed in partnership with the Portsmouth City Council. Each artist was micro-commissioned to produce a poster, guided by the theme 'We Believe—expressions of hope & optimism for our city beyond covid-19.' The trail was artist-led, and they asked artists to produce a piece representing what the theme meant to them. The trail is a 30-min walk around Portsmouth, which can be enjoyed by the local community.

## 4. Conclusions and Recommendations

Creative networks operate as a central node between a wide array of actors, receiving value from the creative network, and at the same time, creating value for the creative network. Positioned at the heart of this system that creates and curates value leading to sustainability, creative networks occupy the role of convenor, communication channel and catalyst, bringing order to an otherwise dispersed creative sector. Creative networks can act as catalysts for sustainability and create a favorable context in which more resilient creative communities can flourish. Our research shows the importance of these networks by highlighting examples as well as the different obstacles and limitations that they face in a creative ecosystem that exists on a continuum between public subsidy and commercialization. In this context, we can summarize our findings as follows:

- Creative networks create value by interconnecting quadruple helix actors: The visualization of the value network shows how creative networks are an anchor point bringing together a variety of actors who, together, create various forms of value for

a wide range of stakeholders. We found that without the creative network being the connector, there would be considerably less connectivity between actors in the local creative ecosystems, which makes these ecosystems less sustainable.

- Creative networks create value in different ways including economic, social and cultural value impacts: Creative networks' direct/indirect and tangible/intangible value-generation mechanisms have an impact on the sustainable development of places, which spills over to the wider economy. The nature of the creative industries—based on small companies and freelancers—makes these networks particularly valuable, with the absence of large players making it unlikely that this will be provided by the sector itself.
- Creative networks face obstacles which hinder their growth including a lack of understanding by all stakeholders: To have a real, long-term sustainable impact, creative networks need to reach a critical mass of actors in their networks. However, they also need to be able to demonstrate the value they bring so they are fully appreciated and understood by all stakeholders, an issue intensified by the current economic environment, where a focus on direct financial value makes this more challenging.

These findings mean we need to continue to develop a narrative and evidence base to explain and measure the value of creative networks across the UK, to better understand how they contribute to the sustainable development of territories. This could be carried out by:

1. Uniting policy efforts and understanding, ongoing resourcing, support and guidance: As key stakeholders in creative networks, further consideration should be given to clarify the role performed by local leadership and other bodies in facilitating networks.
2. Networking the networks: As critical facilitators of the networks, regular meetings for network managers and coordinators will support and enable learning and development. Bringing creative network practitioners together regularly to exchange best practices and learning is vital to knowledge and skills sharing, exchange and collaboration to enhance and develop the overall network ecosystem in the UK [90].
3. Creative network member engagement: It is important to undertake further work to sample network attendees and non-attendees in locations across the UK to better understand the benefits of being involved, and any barriers to engagement to provide additional information to our understanding of value creation for participants, members and users.
4. Supporting research and understanding: Information gathering needs to inform communication strategies to demonstrate the value of networks both to potential members and funders, and to address specific barriers for sustainability. The value generation of creative networks and how this contributes to territorial sustainability—especially in less tangible areas—is still not fully understood but is key for sustainability.

We recommend further research to extend the knowledge base of value networks, especially in relation to the impact of new technologies (e.g., internationalization, group dynamics), participatory forms of knowledge creation (e.g., knowledge leaks through specialized groups such as makers) and fostering other intangible value flows, which we see as the catalyst for sustainable development of regions based on creative industries' activities. Research shows that sustainable-oriented networks share many similarities with the traditional ones [91], yet we need more evidence of how the unicity of these networks turns them into catalysts of environmental, cultural and social sustainability in regions and cities by bringing together specific actors.

**Author Contributions:** The authors made the following contributions: M.K.—investigation and original draft preparation; R.L.—writing (review and editing); S.P.—writing and review; J.L.—editing and review. All authors have read and agreed to the published version of the manuscript.

**Funding:** This study was conducted in conjunction with Cardiff University's Creative Economy Unit, and with Creative Cardiff, a network that connects people working in any creative organisation, busi-

ness or job in the Cardiff region. Find out more via https://www.creativecardiff.org.uk/ (accessed on 20 September 2021).

**Institutional Review Board Statement:** Ethical review and approval were waived for this study, insofar as participants contributed to focus groups in their professional and not personal quality. Furthermore, no personal data was collected through the focus groups and surveys and data was anonymized in the article.

**Informed Consent Statement:** Informed consent was obtained from all subjects involved in the study.

**Data Availability Statement:** The report for the original study on which this article draws can be accessed at: https://creativecardiff.org.uk/joining-dots-research-report-understanding-value-generation-creative-networks (accessed on 20 September 2021).

**Acknowledgments:** This paper and the research behind it would not have been possible without the exceptional support of Kayleigh Mcleod and Vicki Sutton for the organisation of the focus group and the sharing of insights; both granted permission to their names being disclosed in this section.

**Conflicts of Interest:** The authors declare no conflict of interest. The funders had no role in the design of the study; in the collection, analyses, or interpretation of data; in the writing of the manuscript, or in the decision to publish the results.

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
