# Peer review of "Joining the Dots—Understanding the Value Generation of Creative Networks for Sustainability in Local Creative Ecosystems"

_sustainability, doi:10.3390/su132212352_

Round 1

Reviewer 1 Report

This is a very good paper which is very well written and neatly structured. The paper is also well referenced, except for one area. This is the very first paragraph which focuses on creativity giving an overview of the shift in thinking on this topic. There is a large body of research literature that could be referred to as succinctly as was done for networks in the first par of section 2.2. With these additions this would give solid authority to the assertion being made in this first para. This is important as this introductory section is used to justify the paper's concentration on networks. What will be revealed even with a cursory dive into the very extensive literature on creativity is that the push into collective creativity does not just end with distributed creativity, as important as that is, but has moved well beyond that, seeing creativity in terms of creative systems. This is far more compelling for justifying the types of creative networks discussed in the latter part of the paper. As a very cursory introduction to that literature read:

Alexander, V. (2003). Sociology of the arts: exploring fine and popular forms. Malden MA: Blackwell.

Hennessey , B. (2017) ‘Taking a Systems View of Creativity: On the Right Path Toward Understanding’ The Journal of Creative Behavior - Special Issue: In Celebration of the Journal of Creative Behavior’s 50th Anniversary, 51(4), pp. 341–344.

Hennessy, B. & Amabile, T. (2010) ‘Creativity’, Annual Review of Psychology, N61, pp. 569–598

Sawyer, K. (2012) Explaining creativity: The Science of Human Innovation 2nd ed (Oxford: Oxford University Press).

Apart from this inclusion, which is easily completed, I think the paper is certainly well worth publishing.

Reviewer 2 Report

  1. The paper could be better contextualized with respect to the previous and present theoretical background. Some recognized researchers of creative industries (eg. J. Howkins) and creative cities (eg. C. Landry) are even not mentioned. The theoretical problems of creative economy (including classification of the creative sector) and urban creative sustainability could be also better presented.
  2. The paper should include the problems of sustainability.
  3. The self-citation of the author Marlen Komorowski should be better reasoned.

Reviewer 3 Report

The Authors of the paper present an interesting topic of broad cooperation of creative networks, which is very important for dynamic economic development, not only in Great Britain, but also in other countries. Before publication, however, I recommend some changes to the text, namely:

  • please define "ecosystem" in the creative and cultural industries (CCIs);
  • the Authors have taken the topic too broadly: “… creative networks in our study are all: city / town / regional networks; working with multiple creative industries sectors; rooted in and working for a place; in operation for a minimum of 1 year; working for both creative individuals and organizations. " Therefore, I propose to add a justification for the choice;
  • the methodological section should be supplemented and improved

a) "a mixed-method approach including desk research, surveys and the organization of workshops.", where workshops - this is not a method in economic sciences;

b) the methodology does not mention or describe the quadruple helix method, which was announced in the Abstract and is presented (results later in the article) - this should be supplemented;

  • lack of description Figure 2. Authors should explain the assumptions of this diagram and then describe the drawing in general before the detailed description, which is to be found in the further part of the text;
  • lack of the summary - there should be a separate chapter containing the main conclusions and recommendations.
